# A Novel Peptide with Antifungal Activity from Red Swamp Crayfish *Procambarus clarkii*

**DOI:** 10.3390/antibiotics11121792

**Published:** 2022-12-10

**Authors:** Diletta Punginelli, Valentina Catania, Mirella Vazzana, Manuela Mauro, Angelo Spinello, Giampaolo Barone, Giuseppe Barberi, Calogero Fiorica, Maria Vitale, Vincenzo Cunsolo, Rosaria Saletti, Antonella Di Francesco, Vincenzo Arizza, Domenico Schillaci

**Affiliations:** 1Department of Biological, Chemical and Pharmaceutical Sciences and Technologies (STEBICEF), University of Palermo, Via Archirafi 32, 90123 Palermo, Italy; 2Department of Earth and Sea Science (DISTEM), University of Palermo, Viale delle Scienze Blg. 16, 90128 Palermo, Italy; 3Istituto Zooprofilattico Sperimentale della Sicilia “A. Mirri”, Via Marinuzzi, 3, 90129 Palermo, Italy; 4Department of Chemical Sciences, University of Catania, Viale A. Doria 6, 95125 Catania, Italy

**Keywords:** crustacean antimicrobial peptides, antibiotic resistant strains, high-resolution mass spectrometry, antibiofilm activity, *Candida albicans*

## Abstract

The defense system of freshwater crayfish *Procambarus clarkii* as a diversified source of bioactive molecules with antimicrobial properties was studied. Antimicrobial activity of two polypeptide-enriched extracts obtained from hemocytes and hemolymph of *P. clarkii* were assessed against Gram positive (*Staphylococcus aureus*, *Enterococcus faecalis)* and Gram negative (*Pseudomonas aeruginosa*, *Escherichia coli*) bacteria and toward the yeast *Candida albicans*. The two peptide fractions showed interesting MIC values (ranging from 11 to 700 μg/mL) against all tested pathogens. Polypeptide-enriched extracts were further investigated using a high-resolution mass spectrometry and database search and 14 novel peptides were identified. Some peptides and their derivatives were chemically synthesized and tested in vitro against the bacterial and yeast pathogens. The analysis identified a synthetic derivative peptide, which showed an interesting antifungal (MIC and MFC equal to 31.2 μg/mL and 62.5 μg/mL, respectively) and antibiofilm (BIC_50_ equal to 23.2 μg/mL) activities against *Candida albicans* and a low toxicity in human cells.

## 1. Introduction

The increasing failure of many antibiotics to counteract multidrug-resistant pathogens motivates the discovery of novel antimicrobials from unconventional sources [1]. It is known that aquatic invertebrates are important sources of bioactive molecules [2,3,4,5,6]. Among crustaceans, freshwater crayfish represent an interesting source of biological active molecules with potential antimicrobial properties. Freshwater crayfish living in several natural environments (permanent and seasonal rivers, streams and lakes) that are strongly colonized by microorganisms display a great defensive response against invading pathogens to survive and grow in such environments [4]. For this reason, they might represent a significant repository of bioactive molecules with potential antimicrobial features. The defensive ability of crustaceans is based on innate immunity characterized by a low specificity and a prompt reaction time [6]. To control and suppress microbial infections, innate immunity consists mainly of cellular and humoral responses [7]. The cellular response is based on phagocytosis, encapsulation, and hemocyte nodulation [8], whereas humoral immunity relies on the production of reactive intermediates of oxygen and nitrogen, lectins, and antimicrobial peptides (AMPs) [9,10]. As important effectors of the innate immune system, AMPs play a crucial role in defense against pathogenic microorganisms [11]. AMPs are small, amphipathic, gene-encoded oligopeptides (<10 kDa, 10–100 amino acids) with antimicrobial mechanism based on a membrane disruptive attachment or on tackling intracellular targets, causing microbial cell death [12]. Several AMPs have been isolated from crayfish and other crustaceans such as arasin, hyastatin, scygonadin, sphistin, and crustins, which are generally processed in hemocytes and delivered in hemolymph after stimulation [13,14,15].

In this study, we analyzed the antimicrobial activity of polypeptide-enriched extracts of hemocytes and hemolymph from the red swamp crayfish *Procambarus clarkii*. Both extracts were tested against some common bacterial pathogens known for their antibiotic-resistance and against the main nosocomial fungal strain *Candida albicans*.

Fourteen novel peptides were identified and characterized in tested extracts by coupling the nano-reversed phase ultra-high performance liquid chromatography (nanoRP-UHPLC) and the ultra-sensitive high-resolution mass spectrometry (HRMS). Among detected peptides, some potential AMPs were studied through in silico analysis, chemically synthesized, and adopted as platform to obtain further synthetic peptides derivatives, which were tested in vitro and using molecular dynamics (MD) simulations. One of synthetic peptides showed a prominent antifungal and antibiofilm activity against *C. albicans* with a low toxicity toward human cells.

## 2. Results

### 2.1. Antimicrobial Activity and Biofilm Inhibition by P. clarkii Extracts

The antimicrobial properties of polypeptide-enriched extracts achieved from hemocytes and hemolymph of the red swamp crayfish *P. clarkii* were evaluated starting from a 50% *v*/*v* concentration of each sample, against four main bacterial strains: *Staphylococcus aureus* ATCC 25923, *Pseudomonas aeruginosa* ATCC 15442, *Enterococcus faecalis* ATCC 29212, and *Escherichia coli* ATCC 25922, and the yeast *Candida albicans* ATCC 10231.

The results are illustrated in Table 1 and are reported in terms of minimum inhibitory concentration (MIC) expressing the values in percentage *v*/*v* and in concentration μg/mL of protein content of each extract.

A MIC of 50% *v*/*v*, corresponding to 11 μg/mL of protein content, was detected for the hemocytes extract, against all tested pathogens; however, differential MIC values were detected for the hemolymph extract which showed a MIC of 50% *v*/*v*, equivalent to 700 μg/mL of protein content, against the selected bacteria and a MIC of 12.5% *v*/*v*, corresponding to 175 μg/mL of protein content, against the yeast *C. albicans* ATCC 10231.

The interference of biofilm formation by both polypeptide-enriched extract against *S. aureus* ATCC 25923 and *P. aeruginosa* ATCC 15442 was evaluated at sub-MIC concentrations starting from 5% *v*/*v*. The results are reported in terms of biofilm inhibition 50% (BIC_50_), which is the concentration at which an inhibition of 50% of biofilm formation is observed. The analysis showed that the hemocyte polypeptide-enriched extract inhibited biofilm formation of *P. aeruginosa* ATCC 15442 with a BIC_50_ equal to 1 µg/mL, but was inactive against *S. aureus* ATCC 25923, while no inhibition was detected on biofilm of both pathogens with the hemolymph polypeptide-enriched extract (Table 2).

Both polypeptide-enriched extracts exhibiting antimicrobial activity were further investigate with the aim to detect the presence of peptides through nRP-UPLC-High Resolution nESI MS/MS analysis.

### 2.2. MS Analysis of the Amino Acid Sequence of the Peptides Identified in Hemocytes and Hemolymph Extracts

With the aim to identify and characterize peptide components, hemocytes and hemolymph extracts were analyzed by nRP-UPLC-nESI MS/MS. LC-MS/MS data were analyzed by the PEAKS software package that incorporates de novo sequencing results into the database search. The database search did not match any peptide derived from *Crustacea* proteins. However, PEAKS de novo sequencing software allowed us to deduce the primary structure of 14 novel peptides that remained unidentified by the database search algorithm. In particular, 10 novel peptides were identified in the polypeptide-enriched extract from hemocytes, and four in the polypeptide-enriched extract from hemolymph, as shown in Table 3 (MS/MS spectra and de novo interpretation are reported in the Appendix A). Moreover, to predict the potential ability to act as AMPs and to evaluate similarities with known AMPs, the APD3 “Antimicrobial Peptide Calculator and Predictor” tool of the Antimicrobial Peptide Database (APD) was used [16]. A total of eight peptides might be considered potential AMPs due to their ability to associate with bacterial membrane: four from hemocyte extract peptides (#3, #4, #5 and #7) and four (#10, #11, #12 and #14) from hemolymph extract. All results are shown in Table 3. Three peptides derived from the hemocyte extract and one from the hemolymph extract were selected for further analysis to establish the main physico-chemical parameters (Table 4) using the Antimicrobial Peptide Database [16] and Half Life of Peptides, an important tool that allows the prediction of the half-life of peptides in a biological proteolytic environment [17]. Three peptides (peptides #3; #5; #7) identified in the hemocyte polypeptide-enriched extract of *P. clarkii* showed a net positive charge suggesting a more effective role as AMPs compared to anionic peptides [18]. Moreover, peptides #5 and #14 presented with a significant hydrophobic ratio ranging from 40–45% and high values of WimleyWhite whole residues, enhancing their ability to interact and perturb the bacterial membrane, the primary target of AMPs. Other interesting parameters are reported in Table 4.

### 2.3. AMP Predictions through In Silico Analysis

The antimicrobial activity of selected peptides was predicted by using eight available prediction models: the server DPABBs [19] and CellPPD [20,21], to design the antibiofilm and cell-penetrating ability of designed peptides; the Predicted Antigenic Tool [22], to characterize potential presence of antigens; the server HemoPI [23], to predict the hemolytic activity of peptides; ToxinPred [24,25], to characterize potential toxic peptides; Peptide Cutter Server [26], a model useful to predict the presence of cleavage sites; and the server iAMPPred [27], to forecast potential antimicrobial and antifungal activity.

According to the servers CellPPD and DPABBs, all selected peptides do not display the ability to penetrate cell and antibiofilm activity but differential activities for the various peptides were predicted by the in silico analysis, as shown in Table 5.

### 2.4. Antimicrobial Screening of Selected Natural Peptides

The antimicrobial and biofilm inhibition formation assays revealed that chemically synthesized peptides #3, #5, #7, #14, despite their predicted activity by in silico analysis, were not effective in vitro at the maximum tested concentration (MIC > 250 μg/mL) against *S. aureus* ATCC 25923, *P. aeruginosa* ATCC 15442, *E. coli* ATCC 25922, *E. faecalis* ATCC 29212, and *C. albicans* ATCC 10231. To exclude salt-mediated inactivation of AMPs due to conventional medium content, which could interfere with antimicrobial activity, we also determined the half-maximal effective concentration (EC_50_ concentration to inhibit 50% of the viable bacteria), by testing the antimicrobial activity under low-salt conditions in 10 mM sodium phosphate buffer. However, synthetic peptides #3, #5, #7, #14 were not active at the maximum tested concentration of 100 μg/mL (EC_50_ > 100 μg/mL), and antimicrobial activity was not experimentally detected.

### 2.5. Optimization of In Vitro Natural Peptide Potency through Bioinformatic Analysis

With the help of bioinformatics and servers online tools, the natural peptides #3, #5, and #14 were used as a chemical platform to mostly modify three relevant chemical-physical parameters (percentage of hydrophobic ratio, charge, Boman index) to enhance potential antimicrobial and antibiofilm activity. By deletion and/or substitution of amino acid residues, we obtained three new synthetic derivatives with a net positive charge ranging from +1.25 to +3, hydrophobic ratio ≥50%, and Boman Index (the assessment of protein-binding potential affinity) lower than 2.5 kcal/mol (Table 6). Additionally, other significant biological parameters of these peptides were analyzed to define their potential antimicrobial and antibiofilm activity, toxic and hemolytic potential, and the presence of cleavage sites (Table 7). According to online webserver predictions, two synthetic derivative peptides (peptide #3d and peptide #5d) showed high probability (in percentage) to act as good antimicrobial and antifungal peptides compared to peptide #14d, which was predicted to have low antibacterial and antifungal activity (Table 7). Moreover, all three peptides revealed low hemolytic potential, absence of toxicity, and were also characterized by low resistance to proteolysis (Table 7).

### 2.6. Antimicrobial and Antibiofilm Activity of Synthetic Derivative Peptides

The derivative peptides #3d, #5d, and #14d were tested in vitro against four bacterial reference strains *S. aureus* ATCC 25923, *P. aeruginosa* ATCC 15442, *E. coli* ATCC 25922, and *E. faecalis* ATCC 29212; they showed no antibacterial activity at the maximum concentration of 250 μg/mL. However, the synthetic peptide derivative #14d (FHLVWRAGGT) displayed an indicative antifungal activity against *C. albicans* with a MIC value of 31.2 μg/mL and a minimal fungicidal concentration (MFC) of 62.5 μg/mL.

The above mentioned peptides were also analyzed for their ability to interfere with biofilm formation of *S. aureus* and *P. aeruginosa* bacterial strains. None of these peptides showed significant antibiofilm activity at the maximum tested concentration of 200 μg/mL. In contrast, the derivative peptide #14d at a sub-MIC value of 25 μg/mL resulted in effective inhibition of biofilm formation of *C. albicans* ATCC 10231, with a percentage of inhibition value equal to 78.5% (Figure 1).

The value of the biofilm inhibition concentration (BIC_50_) in interfering with biofilm formation was 23.2 μg/mL against *C. albicans* ATCC 10231.

With the aim to investigate on antifungal mechanism of action, three different experiments were performed.

### 2.7. Analysis of C. albicans ATCC 10231 Morphology

Scanning electron microscopy (SEM) was used to analyze potential damage to *C. albicans* ATCC 10231 biofilm after treatment with the synthetic derivative peptide #14d (Figure 2). The treatment of *C. albicans* ATCC 10231 biofilm with peptide #14d at a sub-MIC concentration of 25 μg/mL caused evident variations in the morphology, inducing wrinkles and scars (Figure 2a), compared to the control (Figure 2b).

### 2.8. Effect of Synthetic Peptide #14d on Membrane Integrity

We analyzed whether the fungal membrane integrity was damaged by synthetic peptide #14d using propidium iodide (PI). The PI uptake assay allowed the evaluation of possible damage to *C. albicans* cell membrane through the interaction of the dye with DNA and the release of red fluorescence after the membrane is compromised by antimicrobial agents since healthy membranes are impermeable to PI. We demonstrated that the treatment with synthetic peptide #14d, at sub-MIC concentrations (25 μg/mL, 20 μg/mL, and 15 μg/mL), damaged the cell membrane and induced an increase in red fluorescence intensity (Figure 3A–C). Similarly, the positive control (Sabouraud liquid medium and *C. albicans* ATCC 10231 culture without peptide #14d) did not show any fluorescence because the cell membrane was not damaged (Figure 3D).

ROS overproduction represents another crucial mechanism adopted by the peptides to inhibit biofilm formation. The results obtained in this study suggested that the treatment of *C. albicans* ATCC 10231 with the synthetic peptide #14d did not induce ROS overproduction due to the absence of fluorescence emission (Figure 4).

### 2.9. Citotoxicity Assays on Synthetic Peptide #14d

Cytocompatibility studies of the synthetic peptide #14d performed on HCT-116 (human colon tumor cells) showed that at all tested concentrations the peptide did not interfere with cell viability (viability more than 80% related to the growth control). There were no significant variations between tested concentrations (Figure 5). Moreover, no significant differences in morphology were observed compared to the control.

### 2.10. Molecular Dynamics Simulations

To investigate, at an atomistic level, the structural basis for the antifungal activity shown by the synthetic peptide #14d, we carried out molecular dynamics (MD) simulations on both #14d and the closely related #14 peptide. Their three-dimensional conformations were predicted from their amino acidic sequence using PEP-FOLD3 [28]. Next, to evaluate the stability of the lowest energy models obtained for both peptides, and to relax their structure, 1 μs-long MD simulations were performed in a physiological environment. The most representative structures were then extracted from the MD trajectories using a cluster analysis (Figure 6A). Interestingly, the most relevant conformation of both peptides possesses a similar secondary structural arrangement, mostly showing an α-helical motif. This is confirmed by a more detailed analysis of the peptide’s secondary structure, as evidenced by the Ramachandran plots, in which most of the residues of #14 and #14d show psi and phi angles typical of an α-helical arrangement (Figure 6B).

The electrostatic potential (ESP) of both peptides, calculated using APBS (Adaptive Poisson-Boltzmann Solver) [28] (Figure 6C) showed that peptide #14 had a negative net charge (Table 4) due to the presence of an aspartate residue and, on average, was characterized by a wider distribution of negative values of the ESP (red surface), as compared with peptide #14d. The latter, bearing a net positive charge (Table 6) due to an arginine and a histidine (only partially protonated at physiological pH) residues, on the other hand, clearly showed a prevalence of positively charged regions (blue surface, Figure 6C). In conclusion, the obtained results were attributed the different properties of the two peptides, mainly to their different polarity.

## 3. Discussion

Antibiotic-resistant bacteria represent a major global health problem and a serious medical threat to mankind. In addition, fungal diseases caused by *Candida* species lead to high mortality rates and expensive medical costs for government and hospitalized patients [29,30]. In this scenario, there is an urgent need to discover and develop new therapeutics to overcome microbial resistance [31,32,33]. There is a pressing demand for antimicrobials able to tackle life-threatening biofilm associated infections due to *S. aureus* and *P. aeruginosa*, which are considered priority pathogens for their high or critical antibiotic resistance [34]. *C. albicans* also plays a significant role as nosocomial fungal pathogen and its ability to produce biofilms makes it tolerant to known antifungals [35].

Natural antimicrobial peptides (AMPs) are considered promising alternatives to conventional antibiotics or to act as adjuvants in the treatment of infections. However, they are characterized by some negative features such as toxicity, low resistance to proteolysis, and high cost of isolation and purification. The design of synthetic antimicrobial peptides (SAMPs) represents a different solution to counteract these disadvantages since they show low or no toxicity to mammalian cells and reduced probability to develop antimicrobial resistance [36,37]. Aside from SAMPs, in recent years, further studies have demonstrated the importance of nanozymes as novel therapeutic agents to overcome bacterial and fungal resistance [38], and the application of metal organic frameworks (MOFs) that can be used as bioactive framework materials (BioMOFs) containing antimicrobial agents with improved antibacterial and antifungal activity [39].

In this study, we focused on polypeptide-enriched extracts obtained from hemocytes and hemolymph of *P. clarkii*, which displayed interesting activity against some Gram positive and Gram negative bacteria and the yeast *C. albicans*. By means of nRP-UPLC-nESI MS/MS analyses, we detected the presence of 14 novel peptide sequences (Table 3); among them, the peptides #3, #5, #7, and #14 were similar to host defense peptides identified in other invertebrates. In detail, the peptides #3 and #5 show similarity with AMPs isolated in insects whereas peptide #14 reveals similarity with AMPs derived from amphibians, in particular with temporin, which exhibits antimicrobial activity against Gram positive and Gram negative pathogens and fungi [40].

These natural peptides were chemically synthesized and tested in vitro; however, their predicted in silico activities were not experimentally confirmed although bioinformatic tools and online servers were employed to identify new synthetic derivatives with improved physicochemical characteristics. The major difficulty of employing computational methods to predict synthetic antibiofilm antimicrobial peptides (SAAMPs) is the large number generated, which makes selection of the best sequence for in vitro biological investigation difficult. Among the adopted criteria for the choice of improved sequences to be subjected in vitro analysis, the predicted low toxicity and stability in the host environment are highlighted. Moreover, shorter peptides make chemical synthesis easier and reduce the production costs.

By using in silico tools, we found out that some peptides detected in polypeptide-enriched extracts obtained from *P. clarkii* were a good chemical template to obtain new derivatives with improved biological functions. Of particular importance, the synthetic derivative peptide #14d (FHLVWRAGGTF), selected on the basis of three advantageous chemical-physical parameters (percentage of hydrophobic ratio, charge, Boman index), due to its interesting antifungal activity and low toxicity toward human cells in vitro, may represent a good therapeutic candidate worthy to be developed against the pathogen *C. albicans*. Peptide #14d inhibited the growth of *C. albicans* ATCC 10231 through cell membrane damage causing death [41,42,43]. The peptide showed indicative anticandidal activity, revealing a MIC value equal to 31.2 μg/mL. SEM analysis of *C. albicans* ATCC 10231 planktonic cells treated with peptide #14d showed that the biofilm suffered significant structural damage. In particular, SEM images suggested that peptide #14d caused rupture of the membrane through the formation of scars and cracks, leading to cellular content loss and death. Similar behavior has been reported for two different antifungal peptides, Dermaseptin-S1 [44] and Histatin-5 [45], where the *C. albicans* biofilm suffered severe damage. It has been demonstrated that the main target of SAMPs is the cell membrane and/or cell wall [37]. The major mechanism of SAMPs is the ability to alter cell membrane permeability, reducing the development of resistance by microorganisms [36,37]. To evaluate potential cell membrane damage, fluorescence microscopy analyses were carried out, revealing that peptide #14d caused PI uptake in *C. albicans* ATCC 10231 probably due to membrane pore formation or cell membrane damage. On the contrary, the synthetic peptide #14d did not cause ROS overproduction in *C. albicans* ATCC 10231 biofilm, which are involved in the disruption of essential molecules such as proteins, lipids, and DNA. This result corroborates those reported by Bezerra and collaborators [43], who showed the absence of ROS overproduction by *C. albicans* biofilm after treatments with the conventional antifungal agents nystatin and itraconazole. MD simulations performed on this peptide further suggested that the displayed antifungal activity could be partially related to the overall wide distribution of the positive electrostatic charge in its three-dimensional structure. This, along with the presence of roughly 50% of hydrophobic residues in the sequence (Table 6), creates optimal conditions for the interaction of peptides with target membranes [38]. The synthetic derivative peptide #14d showed antifungal properties similar to some peptides discovered in other crustaceans (Table 8). An antimicrobial peptide with antifungal features has been isolated and characterized previously in freshwater crayfish. Sun et al. (2011) [46] have characterized in *P. clarkii*, an anti-lipopolysaccharide factor (ALF), named PcALF1, that revealed inhibitory activity against *C. albicans*, with a MIC value equal to 20 µg/mL, through specific binding to the microbial polysaccharide β-glucan, the key component of the fungal cell wall. Some significant antifungal peptides have also been isolated from other freshwater crayfish, such as *Pacifastacus leniusculus*, *Cherax quadricarinatus*, and *Eriocheir sinensis*, and exhibit antimicrobial activity at concentrations between 25 and 6 µg/mL. The hemocyanin derived astacidin 1 showed antifungal activity against *C. albicans* and *Trichosporon biegelii*, causing membrane damage via pore formation and membrane depolarization [47]; this mechanism is similar to those observed by Petit et al. [48] in a marine shrimp *Litopenaeus vannamei*. Yu et al. [49] identified a novel type of crustin I from the various organs of the red crayfish *C. quadricarinatus* that exhibited antifungal activity against the fungus *Pichia pastoris*. Moreover, antifungal properties against *P. pastoris* have also been observed in an antimicrobial peptide isolated from *E. sinensis* that showed a strong agglutination and inhibitory activity against the fungus [50].

## 4. Materials and Methods

### 4.1. P. clarkii Collection and Extract Preparation

*P. clarkii* were harvested along a stream in Diga Rosamarina (Palermo, Italy). They were kept in tanks with freshwater and transported to laboratory where they were temporarily maintained in tanks filled with fresh water. The animals were fed pellets and frozen fish ad libitum and were deprived of food for 2 days before the hemolymph sampling. Then hemolymph was taken from ventral sinus using 200 μL of anticoagulant (Nacl 0.14 M, glucose 0.1 M, trysodic citrate 30 mM, citric acid 26 mM, EDTA 10 mM). Hemolymph was centrifuged immediately at 400× *g* for 10 min at 4 °C to isolate the hemocytes. The polypeptide fraction was extracted from hemolymph and hemocytes using acetic acid (2 M) with 1:3 ratio adding antiproteases (1:200). The samples were homogenized, sonicated, and centrifuged again in order to obtain the polypeptide fraction.

### 4.2. Protein Concentration

The protein concentration of the sample was measured using the Bradford method [51], and evaluated at an absorbance of 595 nm using a spectrophotometer.

### 4.3. Bacterial Strains

Four reference bacterial strains and one fungal strain were used: *S. aureus* ATCC 25923, *P. aeruginosa* ATCC 15442, *E. faecalis* ATCC 29212, *E. coli* ATCC 25922, and *C. albicans* ATCC 10231. The media used in this study were Tryptic Soy Broth (TSB, Sigma-Aldrich, Merck-Life Sciences S.r.l., Milano, Italy), Tryptic Soy Agar (TSA), Mueller Hinton II (Sigma-Aldrich, Merck-Life Sciences S.r.l., Milano, Italy), and Sabouraud medium for the yeast strain.

### 4.4. Determination of Minimal Inhibitory Concentrations (MICs)

Natural extracts derived from hemocytes and hemolymph of *P. clarkii* were liophylized and then resuspended in ultrapure water to obtain a protein content of 22 µg/mL and 1400 µg/mL, respectively. MIC values of extracts were determined by a micro-method previously described [52] based on serial dilution 1:2 of the extracts, using Mueller Hinton II medium and Sabouraud medium (for *C. albicans*), in 96-well microplates and starting from concentrations of 50% *v*/*v* to 1.5 % *v*/*v* in a final volume equal to 100 μL. Thus, 10 μL of microbial suspension (5 × 10^6^ CFU/ mL) obtained from a bacterial culture grown at 37 °C for 24 h on Tryptic Soy Agar (TSA) in 5 mL of 0.9% NaCl, with a turbidity equivalent to a 0.5 McFarland standard was added to 96-well plates and incubated at 37 °C for 24 h. MICs of extracts were determined by a microplate reader (GloMax^®^-Multi Detection System, Promega Italia s.r.l, Milan, Italy) with the lowest concentration of the sample whose optical density (OD) at 570 nm was similar to negative control wells (broth without inoculum to control the sterility of medium). Moreover, a positive growth control (bacterial tested strain in the medium without extracts to compare the growth of cells with samples) and a sample control (extract solutions without inoculum to control the absorbance of the samples at different concentrations) were also added.

The same methods and media were used to determine the antimicrobial activity of synthetic peptides #3, #5, #7, and #14, and derivatives. They were dissolved in ultrapure water, or 1 mL ultrapure water with 40 μL NH_4_OH (28% *v*/*v*) in the case of peptide #14, to obtain a stock solution of 5 mg/mL. Working solutions of 250 μg/mL were prepared for each peptide in MHII or Sabouraud, and test solutions in a 96 wells plate were obtained by diluting in a series 1:2. Each assay was performed in triplicate [53].

### 4.5. Determination of EC50

The antimicrobial activity of the chemically synthesized peptides #3, #5, #7, and #14, and their derivatives, against selected strains were evaluated in terms of peptide half-maximal effective concentration to inhibit 50% of the viable bacteria (EC50), as reported by other authors [54]. Briefly, 5 × 10^4^ CFU per well of tested microorganisms were incubated with peptide concentrations from 100 to 0.15 μg/mL (1:2 serial dilutions) for 3 h at 37 °C in 100 μL of 10 mM sterile sodium phosphate buffer. After 3 h, 10 μL of each dilution was spotted in triplicate on Tryptic Soy Agar plates and incubated at 37 °C for 24 h. After the incubation time, the number of colonies from any sample were counted and compared with the CFU viable count of the none-treated growth control.

### 4.6. Inhibition of Biofilm Formation (Crystal Violet Method)

Sub-MIC concentrations were tested for their ability to interfere with the growth as sessile community of *S. aureus* ATCC 25923, *P. aeruginosa* ATCC 15442, and *C. albicans* ATCC 10231, as previously described [55]. Bacterial strains or the fungal strain were incubated in test tubes with TSB or Sabouraud (5 mL) containing 2% *w*/*v* glucose at 37 °C for 24 h. After that, small aliquots (2.5 μL) from the diluted microbial suspensions whose turbidity was equivalent to a 0.5 McFarland standard were added to each well of a polystyrene sterile flat-bottom 96-well plate filled with TSB or BS (200 μL) with 2% *w*/*v* glucose [56,57]. Sub-MIC concentration values of amino acidic fraction from hemocytes and hemolymph of *P. clarkii* or of chemically synthesized peptides were directly added to the wells and the plates were incubated at 37 °C for 24 h. After biofilm growth, samples were washed twice with sterile 0.9% NaCl and stained with 200 μL of 0.1% *w*/*v* crystal violet solution for 15 min at 37 °C [58]. Excess solution was discarded, and the plate was washed twice using tap water. Then, a volume of 200 μL of ethanol was added to each stained well to solubilize the dye. Last, the absorbance was determined at 600 nm using a microplate reader (GloMax^®^-Multi Detection System, Promega Italia s.r.l, Milan, Italy) [59]. The experiments were run at least in triplicate, and three independent experiments were performed. The percentage of inhibition was calculated through the formula:% Inhibition = [(OD growth − OD sample)/OD growth control] × 100.

BIC_50_ (the concentration at which the percentage of inhibition of biofilm formation is equal to 50%) was calculated using AAT Bioquest, Inc. Quest Graph™ IC50 Calculator (v.1), retrieved from https://www.aatbio.com/tools/ic50-calculator-v1 (accessed on 3 March 2022) [60].

### 4.7. Mass Spectrometry Analysis

Mass spectrometry analysis of the two polypeptide-enriched extracts obtained from hemocytes and hemolymph of *P. clarkii*, was carried out by nHPLC-nESI MS/MS using a Dionex UltiMate 3000 RSLCnano system coupled on-line with an Orbitrap Fusion Tribrid^®^ (Q-OT-qIT) mass spectrometer (Thermo Fisher Scientific, Bremen, Germany). MS data, obtained using 1 μL (corresponding to 25 ng) of each solution, were achieved as previously reported [53]. The full scan mass spectra (*m*/*z* range: 200–1600) were acquired in high resolution mode (Mass Resolution 120K, @ 200 *m*/*z*) by an Orbitrap analyzer, whereas the tandem mass spectra (MS/MS data) were achieved in low resolution mode using a linear trap analyzer.

### 4.8. Database Search

MS/MS data were analyzed by the PEAKS de novo sequencing software (v. 10.0, Bioinformatics Solutions Inc., Waterloo, ON, Canada) and searched against the UniProt protein sequences database limited to *Crustacea* taxonomy (442 entries, March 2020 release), using the same parameters reported in [53]. All Peptides Spectrum Matches were also manually checked.

### 4.9. AMP Prediction and Bioinformatic Analysis

Physicochemical parameters of detected sequences and synthetic peptides were obtained employing the “APD3: Antimicrobial Peptide Calculator and Predictor” tool of the Antimicrobial Peptide Database (APD) [16]. The predicted half-life and stability of the sequences in an intestine-like proteolytic environment were evaluated using “HLP: Web server for predicting half-life of peptides in intestine-like environment” [17]. To describe biological properties of the peptides, we analyzed the sequences through online free servers: (1) dPABBs (http://abopenlab.csir.res.in/abp/antibiofilm/protein.php, accessed on 20 November 2022); (2) CellPPD (http://crdd.osdd.n14valuatava/cellppd/, accessed on 20 November 2022), which is used to design cell-penetrating peptides. The antibacterial and antifungal potential were assessed using the iAMPpred tool (http://cabgrid.res.in:8080/amppred/, accessed on 20 November 2022) whereas hemolytic and toxic activities were predicted using HemoPI (https://webs.iiitd.edu.in/raghava/hemopi/, accessed on 21 November 2022) and ToxinPred tools (http://crdd.osdd.net/raghava/toxinpred/, accessed on 21 November 2022), respectively. Moreover, the allergic potential was calculated through the antigenic prediction tool (http://imed.med.ucm.es/Tools/antigenic.pl9, accessed on 21 November 2022). The pI (https://web.expasy.org/protparam/protpar-ref.html, accessed on 22 November 2022), the presence of cleavage sites (https://web.expasy.org/peptide_cutter/, accessed on 22 November 2022), molecular mass (http://aps.unmc.edu/AP/, accessed on 22 November 2022) and resistance to proteolysis (http://crdd.osdd.net/aluatava/hlp/help.html, accessed on) were also determined [37].

### 4.10. Peptide Synthesis

The tested peptides were synthesized by GenScript Biotech (Leiden, Netherlands), according to our design and indications. The peptides were obtained using fluorenymethyloxycarbonyl protecting group (Fmoc) solid phase technology. The quality and purity of each peptide (≈98%) was determined through mass spectrometry (MS) analyses and high-performance liquid chromatography. The powdered peptides were kept at –20 °C for storage.

### 4.11. Molecular Dynamics (MD) Simulations

The PEP-FOLD3 webserver [61] was employed to obtain a folded structure of the peptide #14 and its derivative #14d. Namely, for each structure, 100 runs were performed, and clusters were sorted by sOPEP energy. This software predicts, using the Hidden Markov Model-derived structural alphabet, a limited number of local conformations. Afterwards, these are assembled using a greedy procedure employing a coarse-grained energy function. The lowest energy models were then taken as a starting point for the following simulations (Figure 3). In particular, the stability of the peptides in physiological conditions was assessed by molecular dynamics (MD) simulations, following a previously reported procedure [62,63]

MD simulations were performed using GROMACS 2020.2 [64] along with the Amber99SB-ILDN force field [65]. The two peptides, #14 and #14d, were solvated in a cubic box having a distance between the walls and the solute atoms of 12 Å and filled with TIP3P water molecules. The resulting systems were then slowly heated in 10 ns to achieve a final temperature of 300 K using a Langevin thermostat. The pression control (1 atm) was performed using a Berendsen barostat [66]. Finally, production NPT runs of 1 μs were performed for each peptide, for a cumulative simulation time of 2 μs.

Figures and plots were obtained by the VMD software [67]. Clustering analysis was performed using a hierarchical agglomerative approach via the gmx cluster tool [67]. Ramachandran plots were calculated using the gmx rama tool. Finally, electrostatic potential was calculated using APBS (Adaptive Poisson-Boltzmann Solver) calculations [28].

### 4.12. Cytocompatibility Assays

The studies were performed on HCT-116 (human colon tumor cells) using the MTS colorimetric assay. Cells were cultured in Dulbecco’s Minimum Essential Medium (DMEM) supplemented with fetal bovine serum (10% *v*/*v*), penicillin-streptomycin solution (1% *v*/*v*), glutamine solution (1% *v*/*v*), and amphotericin B solution (1% *v*/*v*), and were maintained at 37 °C and 5% of CO_2_ with a humidified atmosphere. After trypsinization, cells were counted, re-suspended in DMEM, and seeded into 96-well culture plates at a density of 1.0 × 10^4^ cells for well and incubated overnight to allow adhesion. Then, different concentrations of the derivative peptide #14d (1, 0.5, 0.25, 0.125, 0.0625, and 0.03125 mg/mL) were prepared by dispersing the sterile peptide into DMEM and adding 100 μL for well to the cell layer. After 24 and 48 h of cell incubation, cell viability was evaluated using the MTS assay, and the absorbance was detected at 492 nm using a UV-vis absorbance microplate reader. Cell viability was expressed as a percentage of viability compared to growth control. Experiments were performed in triplicate.

### 4.13. Cell Membrane Integrity Assay

The cell membrane integrity of *C. albicans* ATCC 10231 after exposure to peptide #14d was evaluated using the PI uptake assay, as described by Bezerra et al. [43], with some modifications. *C. albicans* ATCC 10231 was incubated with the peptide #14d in the same conditions reported for inhibition of biofilm formation. Thus, 20 μL of propidium iodide (PI, Molecular probes^TM^, Fisher Scientific, Erembodegem, Belgium) was added to each well and the plate was incubated in the dark for 30 min at 25 °C. Then, the plate was washed three times with 0.9% NaCl and observed with an Olympus FV1200 confocal microscope with TIRF (Total internal reflection fluorescence) integrated, using an excitation wavelength of 535 nm and emission wavelength of 617 nm.

### 4.14. Overproduction of Reactive Oxygen Species (ROS)

The ROS overproduction in *C. albicans* ATCC 10231 due to peptide #14d was evaluated using the method described by Bezerra et al. [43], with some modifications. *C. albicans* ATCC 10231 was incubated with peptide #14d in the same conditions reported for biofilm inhibition formation. Then, the formed biofilm was washed with 0.9% NaCl three times to remove the Sabouraud liquid medium. After this procedure, 20 μL of 2′,7′ dichlorofluorescein diacetate (DCFH-DA) was added to each well and the plate was incubated in the dark for 30 min at 25 °C. DCFH-DA is oxidized by ROS to dichlorofluorescein, which emits green fluorescence [68]. Lastly, the biofilms were washed with 0.9% NaCl three times and observed with a fluorescence microscope with an excitation wavelength of 488 nm and emission wavelength of 525 nm.

### 4.15. Scanning Electron Microscopy (SEM)

The morphological changes in *C. albicans* ATCC 10231 cells were assessed by scanning electron microscopy (SEM, Phenom ProX, PhenomWorld, Eindhoven, Netherlands), using the method described by Bezerra et al. [43], with some modifications. Biofilms were fixed with a 4% formaldehyde solution in DPBS for 30 min at 37 °C. Then, the biofilms were washed three times with sodium phosphate buffer (pH = 7.4). Next, samples were dehydrated with increased ethanol concentrations (30%, 50%, 70%, and 100%) for 10 min at 24 °C. Lastly, the final dehydration was conducted with hexamethyldisilane (HDMS) for 10 min, removing the solvent at the end of washing to allow the samples to air dry. The samples were analyzed by scanning electron microscopy (SEM, Phenom ProX, PhenomWorld).

## 5. Conclusions

Novel peptides from *P. clarkii* could be useful scaffolds in the advancement of research on antimicrobial, antibiofilm, and antifungal peptides. A promising anticandidal synthetic peptide with interesting antibiofilm activity and low toxicity toward human cells (viability more than 80%) has been obtained in this study.

## Figures and Tables

**Figure 1 antibiotics-11-01792-f001:**
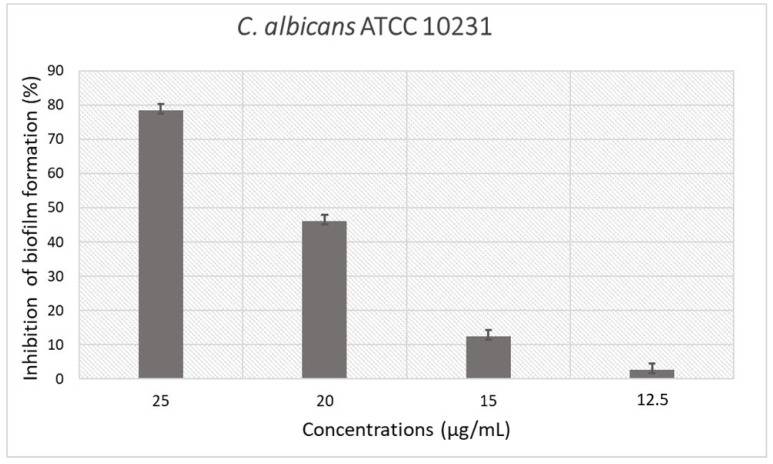
Inhibition of biofilm formation of the derivative peptide #14d against *C. albicans* ATCC 10231.

**Figure 2 antibiotics-11-01792-f002:**
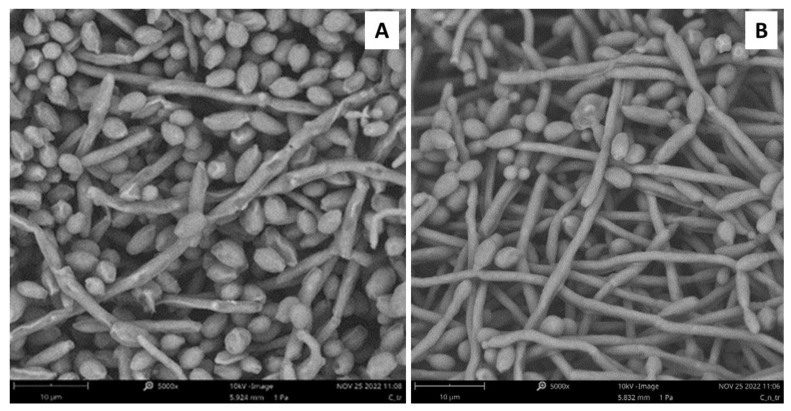
SEM images showing *C. albicans* ATCC 10231 biofilm after 24 h of treatment with synthetic derivative peptide #14d at sub-MIC concentration of 25 μg/mL. (**A**), and non-treated control (**B**). Bars: 10 μm.

**Figure 3 antibiotics-11-01792-f003:**
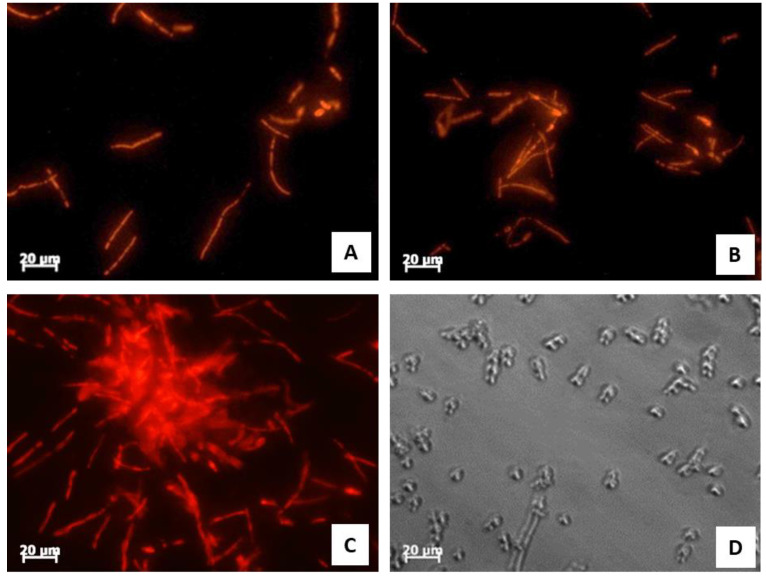
Characterization of membrane damages caused by peptide #14d in *C. albicans* ATCC 10231 analyzed by fluorescence microscope (40×) using propidium iodide (PI). Cells were treated with increasing concentrations of peptide #14d (15–25 μg/mL) and incubated for 24 h.; 25 μg/mL (**A**), 20 μg/mL (**B**), and 15 μg/mL (**C**). Growth control treated of *C. albicans* ATCC 10231 in Sabouroud medium (**D**). Bars: 20 μm.

**Figure 4 antibiotics-11-01792-f004:**
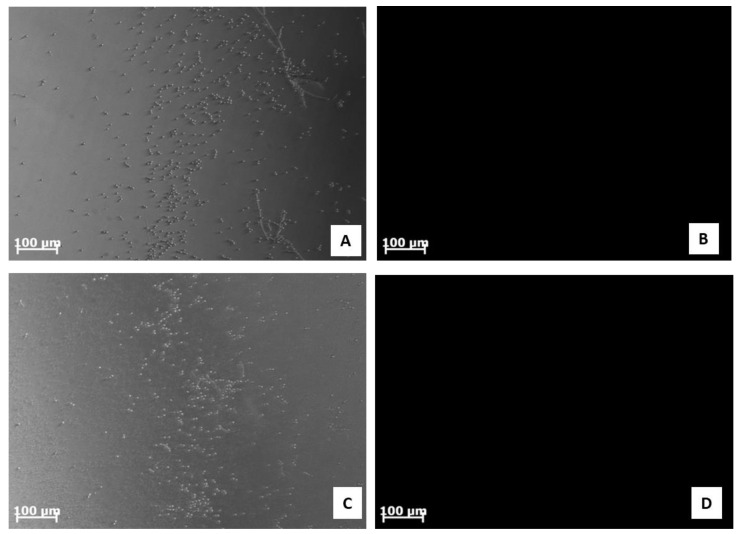
Fluorescence images illustrating ROS overproduction on inhibition of biofilm of *C. albicans* ATCC 10231 cells. ROS production was detected using 2′,7′ dichlorofluorescein diacetate (DCFH-DA). (**A**,**B**) *C. albicans* ATCC 10231 cells treated with peptide #14d at 25 μg/mL; (**C**,**D**) *C. albicans* ATCC 10231 cells treated with peptide #14d at 20 μg/mL. Bars: 100 μm.

**Figure 5 antibiotics-11-01792-f005:**
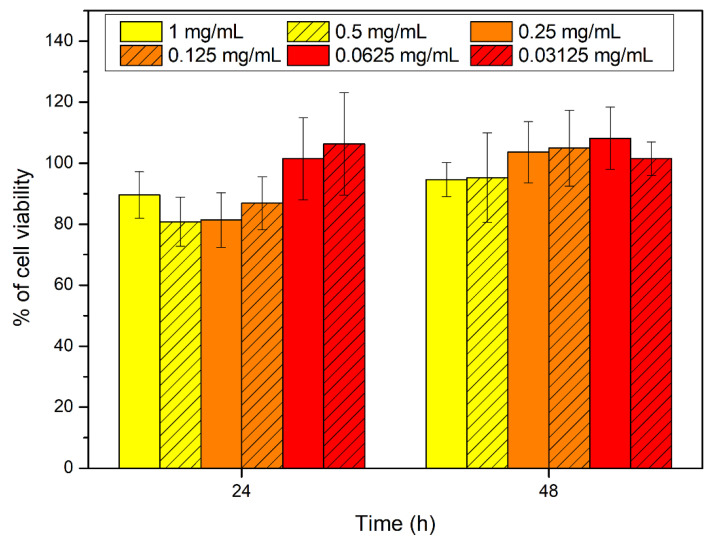
Viability of HCT-116 in presence of peptide #14d after 24 h and 48 h of incubation.

**Figure 6 antibiotics-11-01792-f006:**
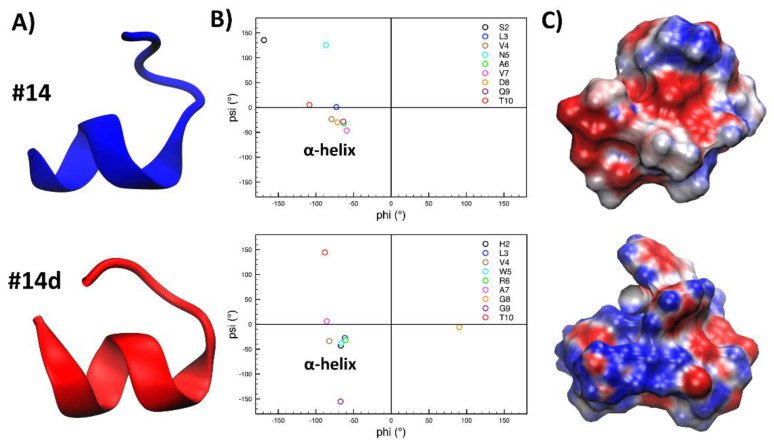
(**A**) Cartoon representation of the structure of the most representative cluster of peptides #14 (blue) and #14d (red), obtained from MD simulations. (**B**) Ramachandran plots showing the values of the psi and phi angles assumed by each residue. (**C**) Electrostatic potential (red and blue for negative and positive values, respectively) for peptides structures, shown as surfaces.

**Table 1 antibiotics-11-01792-t001:** Minimum inhibitory concentration (MIC) of polypeptide-enriched extracts from hemocytes (protein content of extract 22 μg/mL) and hemolymph (protein content of extract 1400 μg/mL) of *P. clarkii*. Activity expressed as MIC in % *v*/*v* or μg/mL of protein content in brackets.

	MIC	
	Hemocytes	Hemolymph
*S. aureus* ATCC 25923	50% *v*/*v* (11 μg/mL)	50% *v*/*v* (700 μg/mL)
*P. aeruginosa* ATCC 15442	50% *v*/*v* (11 μg/mL)	50% *v*/*v* (700 μg/mL)
*E. coli* ATCC 25922	50% *v*/*v* (11 μg/mL)	50% *v*/*v* (700 μg/mL)
*E. faecalis* ATCC 29212	50% *v*/*v* (11 μg/mL)	50% *v*/*v* (700 μg/mL)
*C. albicans* ATCC 10231	50% *v*/*v* (11 μg/mL)	12.5% *v*/*v* (175 μg/mL)

**Table 2 antibiotics-11-01792-t002:** Biofilm Inhibition concentration 50% (BIC_50_) in µg/mL.

	BIC_50_	
	Hemocytes	Hemolymph
*S. aureus* ATCC 25923	>1	>70
*P. aeruginosa* ATCC 15442	1	>70

**Table 3 antibiotics-11-01792-t003:** Identified peptides of *P. clarkii* from hemocytes and hemolymph polypeptide-enriched extracts, and some characteristics as potential AMPs.

Polypeptide-Enriched Extracts	#No.	Identified Peptides	Better Chance to Be an AMP Predicted Ability to Interact with Membranes	Percentages of Similarity with Already Described AMPs
Hemocytes	#1	MFLHGHAV ^a^	-	44.44% plicatamide (marine tunicate, invertebrates)
	#2	EGLDDDERL ^b^	-	44.44% SAAP fraction (mammals)
	#3	SSGYGGYGGRF	yes	50% Crinicepsin I (insects)
	#4	LNVQAQMLLQ	yes	38.46% Temporin–1Ee (frogs, amphibians)
	#5	NNWTGADCKAATLK	yes	42.85% Panurgirn I (insects)
	#6	SHGDSALSSTF	-	42.86% Temporin-1DYa (frogs, amphibians)
	#7	YGGYFGNR	yes	50% Crinicepsin I (insects)
	#8	ETEASLTAALPRW	-	38.46% RP9 (reptiles)
	#9	LVDSNGALLDELPVAR	-	41.18% Peptide #4 (frogs, amphibians)
	#10	KLLLDNSAEDLEELASHK	yes	45% Hb 98–114 (mammals)
Hemolymph	#11	AADSFGETFAATL	yes	38.46% Urechistakynin II (marine worms, invertebrates)
	#12	FDTLSSHLVATD	yes	42.86 Temporin-SN4 (frogs, amphibians)
	#13	ETAPLSGVCF	-	41.67% Peptide 7 (molluscs, marine invertebrates)
	#14	FSLVNAVDQTT	yes	38.46% VmCT2 (scorpions, arthropds)

^a^ The N-terminal methionine was in oxidized form; ^b^ The N-terminal glutamic acid residue was in the pyroglutamic acid form.

**Table 4 antibiotics-11-01792-t004:** Principal physicochemical parameters of potential AMPs from hemocytes and hemolymph polypeptide-enriched extracts.

	Peptide #3	Peptide #5	Peptide #7	Peptide #14
Peptide sequence	SSGYGGYGGGRF	NNWTGADCKAATLK	YGGYFGNR	FSLVNAVDQTT
Monoisotopic Theoretical mass (Da)	1163.499	1491.714	932.414	1193.592
Net charge	+1	+1	+1	−1
Isoelectric point	6.228	6.091	6.360	5.474
Wimley–White whole–residue (kcal/mol)	−1.88 kcal/mol	2.2 kcal/mol	−1.75 kcal/mol	1.26 kcal/mol
Hydrophobic ratio (%)	8	40	13	45
Hydrophobic moment (µH)	0.336	0.044	-	0.315
Protein–binding potential Boman index (kcal/mol)	1.11 kcal/mol	1.66 kcal/mol	2 kcal/mol	1.05 kcal/mol
Half-life (s)	1.098	0.733	2.198	0.304
Stability in a biological proteolytic environment	high	normal	high	normal

**Table 5 antibiotics-11-01792-t005:** In silico predicted properties by using the following tools and servers, dPABBs (http://ab-openlab.csir.res.in/abp/antibiofilm/protein.php, accessed on 20 November 2022); CellPPD (http://crdd.osdd.net/raghava/cellppd/submission.php, accessed on 20 November 2022).

Properties	Peptide #3	Peptide #5	Peptide #7	Peptide #14
Sequences	SSGYGGYGGRF	NNWTGADCKAATLK	YGGYFGNR	FSLVNADQTT
CPP (cell penetrating peptides)	Not CPP	Not CPP	Not CPP	Not CPP
Antibacterial activity	68.9%	86.4%	49,3%	59%
Antifungal activity	71.5%	88.2%	65.2%	48%
Antibiofilm activity	Not predicted	Not predicted	Not predicted	Not predicted
Allergic potential	No	Yes	No	No
Hemolytic potential (probability)	0.49	0.50	0.49	0.49
Toxicity	No	No	No	No
Degradation by trypsin	Yes	Yes	Yes	No
Degradation by pepsin (pH = 1.3)	Yes	Yes	Yes	Yes
Degradation by pepsin (pH > 2)	Yes	Yes	Yes	Yes

**Table 6 antibiotics-11-01792-t006:** Physicochemical parameters of selected synthetic derivative peptides.

Starting Natural Peptide Sequence	Derivative Synthetic Peptide	Derivative Synthetic Peptide Sequence	Monoisotopic Theoretical Mass (Da)	Net Charge	Boman Index (kcal/mol)	Hydrophobic Ratio (%)
#3 SSGYGGYGGRF	Pep #3d	IIIRKGRW	1041.31	+3	2.17	50
#5 NWTGADCKAATLK	Pep #5d	NWWTGARCKAATLK	1634.87	+3	1.46	50
#14 FSLVNADQTT	Pep #14d	FHLVWRAGGTF	1290.49	+1.25	0.11	55

**Table 7 antibiotics-11-01792-t007:** Potential biological parameters of synthetic derivative peptides.

Peptide	Antibiofilm Activity ^1^	CPP ^2^	Antimicrobial Activity ^3^	Antifungal Activity ^3^	Hemolytic Potential ^4^	Toxic Potential ^5^	Trypsin Cleavage Site	Pepsin Cleavage Site pH 1.3–pH > 2
Peptide #3d	Yes	No	82%	69%	0.49	No	Yes	No-Yes
Peptide #5d	Yes	Yes	91%	61%	0.48	No	Yes	Yes-Yes
Peptide #14d	Yes	No	41%	14%	0.48	No	Yes	Yes-Yes

^1^ The antibiofilm activity was evaluated using dPABBs tool (https://ab-openlab.csir.res.in/abp/antibiofilm/, accessed on 20 November 2022); ^2^ CPP (Cell Penetrating Peptide) was assessed through the CellPPD tool; ^3^ The antimicrobial and antifungal properties were calculated using iAMPpred tool (http://cabgrid.res.in:8080/amppred/, accessed on 20 November 2022); ^4^ The hemolytic potential was evaluated through HemoPI tool (https://webs.iiitd.edu.in/raghava/hemopi/, accessed on 21 November 2022); ^5^ The toxin potential was determined using ToxinPred tool (https://webs.iiitd.edu.in/raghava/toxinpred/, accessed on 21 November 2022). The cleavage sites were examined through the Peptide Cutter (https://web.expasy.org/peptide_cutter/, accessed on 20 November 2022).

**Table 8 antibiotics-11-01792-t008:** Antifungal peptides isolated freshwater crayfish.

Peptide	Source	Target Fungi	In Vitro MIC (μg/mL)	References
PcALF1	*Procambarus clarkii*	*C. albicans*	20.0	[46]
Astacidin 1	*Pacifastacus leniusculus*	*C. albicans* *T. biegelii* *Malassezia furfur* *Trichophyton rubrum*	6.3 6.3 12.5 25.0	[47,48]
Crustin I	*Cherax* *quadricarinatus*	*C. albicans*	20.0	[49]
Es-DWD1	*Eriocheir sinensis*	*P. pastoris*	30.0	[50]

## Data Availability

The data presented in this study are available on request from the corresponding author.

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
