# Peer review of "A Novel Peptide with Antifungal Activity from Red Swamp Crayfish Procambarus clarkii"

_antibiotics, 2022, doi:10.3390/antibiotics11121792_

Round 1

Reviewer 1 Report

 The authors reported the antifungal of peptides in red swamp crayfish Procambarus clarkia. The submission can be accepted after revision, taking into account the following points:-

 1.       The title should be revised to be clear, precise, short, and informative. Redundant words such as ‘A promising’,  and ‘synthetic peptide from novel’ should be removed.

2.      The full characterization of the peptide should be included using routine analysis techniques.

3.      A comparison with previously published antifungal peptides should be discussed and summarized in a Table.

4.      The peptide degradation should be included.

5.      The antifungal mechanism should be discussed and supported with experimental data.

6.      References should be updated. I suggest these References; DOI: 10.1039/D0TB00894J; https://doi.org/10.1002/wnan.1769; https://doi.org/10.3390/app12010260;

7.      The language should be revised, and typos should be corrected.

Author Response

  1. The title should be revised to be clear, precise, short, and informative. Redundant words such as ‘A promising’, and ‘synthetic peptide from novel’ should be removed.
  • Reply: Thank you for the suggestion. We have revised and replaced the title with: A novel peptide with antifungal activity from red swamp crayfish Procambarus clarkii.

  1. The full characterization of the peptide should be included using routine analysis techniques.
  • Reply: Thank you for the comment. We purchased synthetic peptides (designed by us) from a specialized company (GenScript) . The company guarantees high-level quality control processes. In particular, each peptide has been triple checked for quality via both mass spectrometry (MS) and high performance liquid chromatography (HPLC) analyses after each step during peptide purification and quality control (QC) procedures.

  1. A comparison with previously published antifungal peptides should be discussed and summarized in a Table.
  • Reply: Thank you for the suggestion. We agree to strengthen this point and added new information about a comparison with previously published antifungal peptides. We have summarized previously published antifungal peptides in new table (Tab. 8) and inserted new sentences to introduce the topic in paragraph discussion (367-385)

  1. The peptide degradation should be included.
  • Reply: Thank you for the suggestion. We have reported “predicted” peptide degradation using in silico tool and have inserted the information in table 7. Though speculative, these aspects will be investigated in future experiments to understand better. Morover, we added others information on significant biological parameters of peptides in the paragraph 2.4 (lines 174-182), summarizing in a new table their properties (Table 7).

  1. The antifungal mechanism should be discussed and supported with experimental data.
  • Reply: Thank you for the suggestion. We have performed a new set of experiments to elucidate the antifungal mechanism of the peptide, evaluating the effect of peptide on membrane integrity. In particular, we added in Materials and Methods the following assays: Cell membrane integrity assay, Overproduction of Reactive Oxygen Species (ROS), Scanning Electron Microscopy (SEM) ( Lines 598-630). We added a part in the discussion dedicated to new results obtained (Lines 344-362). In general, AMPs can interact with a variety of microbial targets and new experiments will be performed in the future for a more in-depth study.

  1. References should be updated. I suggest these References; DOI: 10.1039/D0TB00894J; https://doi.org/10.1002/wnan.1769; https://doi.org/10.3390/app12010260;
  • Reply: Thank you for the suggestion. We have updated the references in the paragraph “discussion” (Line 315, line317) .

  1. The language should be revised, and typos should be corrected.
  • Reply: Thanks for the suggestion. We have corrected the typos and the language was revised by a native speaker.

Reviewer 2 Report

The article entiltled "Apromising antifungal synthetic peptide from novel peptides in red swamp crayfish Procambarus clarkii" is written according to journal style. However, a few minor correction are required.

1. Please check line 32-the citation style [2-6]

2. table 2- foot note......pyroglutamic acid fo -> form

3. line 156- in vitro it can be in italics 

4.Line 307 -: How the Initial sample was prepared and subsequently diluted. what was the activity of Negative and positive controls used in the study. What was the activity of diluent used in the preparation of extract (initial step).

5. Conclusion : "low toxicity toward human cells" explain in terms of statistical analysis

6. Plagiarism is 38%. Please reduce especially the highlighted text. 

Author Response

  1. Please check line 32-the citation style [2-6]
  • Reply: Thank you for the suggestion. We have corrected the citation style.

  1. table 2- foot note......pyroglutamic acid fo -> form
  • Reply: Thank you for the suggestion. We have corrected the word.

  1. line 156- in vitro it can be in italics
  • Reply: Thank you for the suggestion. We have corrected the mistake (Line 158).

  1. Line 307 -: How the Initial sample was prepared and subsequently diluted. what was the activity of Negative and positive controls used in the study. What was the activity of diluent used in the preparation of extract (initial step).

  • Reply: Thanks for your suggestions, we added one sentence in the paragraph MEM (lines 414-416), the procedures used for polypeptide fraction extraction from clarkii is described in paragraph 4.1. Considering your comment: “what was the activity of Negative and positive controls used in the study” we explain in lines 428-431  that negative control was broth without microbial inoculum, and positive growth control was bacterial tested strain in the medium without extracts, so no antimicrobial activity is detected. Considering your comment: What was the activity of diluent used in the preparation of extract (initial step)”, we use as a diluent ultrapure water, so no antimicrobial activity is detected.

  1. Conclusion: "low toxicity toward human cells" explain in terms of statistical analysis

Answer: Thank you for the suggestion.

  • Reply: Thanks for your comments. We indicated in the results paragraph that there are no significant variations between the tested concentrations, consequently since the vitality is 80% higher than the control (especially at 48h), for this reason, we do not believe it is useful to report statistical analysis.

  1. Plagiarism is 38%. Please reduce especially the highlighted text.
  • Reply: Thank you for the suggestion. We have changed many highlighted sentences reported with a high percentage of plagiarism. However, we have observed that claims of potential plagiarism often refer to methodological parts and therefore it is not easy to describe in a very different way.